# Vaping: The Key to Quitting Cigarettes or a Gateway to Addiction?

**DOI:** 10.3390/medicina60091541

**Published:** 2024-09-20

**Authors:** Jessica Emily Hill, Marepalli Bhaskara Rao, Tianyuan Guan

**Affiliations:** 1College of Public Health, Kent State University, Kent, OH 44240, USA; 2Department of Biostatistics, Health Informatics & Data Sciences, University of Cincinnati, Cincinnati, OH 45221, USA; raomb@ucmail.uc.edu

**Keywords:** vaping, cigarette smoking, adolescents, depression, logistic regression

## Abstract

*Background and Objectives*: In 2022, the Centers for Disease Control and Prevention (CDC) discovered that 2.55 million middle and high school students were using electronic cigarettes (e-cigarettes) in the US alone. E-cigarettes allow people to inhale a nicotine solution (e-liquid) into their bodies. While e-cigarettes are often advertised as a way to help people quit traditional tobacco products, the World Health Organization (WHO) has stated that there is no proof of e-cigarettes being effective at reducing an individual’s tobacco use. The objective of this study is to determine why adolescents start smoking e-cigarettes. *Materials and Methods*: For the following study, the National Youth Tobacco Survey (NYTS) 2021 was used. This is a nationally representative cross-sectional survey that includes middle and high school students. The data were analyzed using SAS v.9.4. The odds ratio for smoking e-cigarettes was evaluated for both sex and race via a logistic regression. *Results*: It was discovered that the percentage of only e-cigarette smokers (12.10%) was 5.5 times the percentage of only traditional smokers (2.19%). Additionally, the median age at which adolescents started smoking e-cigarettes was fifteen years with an IQR (Inter Quartile Range) of two. We used a logistic regression to show that biological sex and race were significant predictors of e-cigarette usage. *Conclusions*: In general, we saw that adolescents were mostly starting to smoke e-cigarettes because their friends were doing it, out of curiosity, they were depressed or anxious, and/or to get a “high”, implying that adolescents mostly started smoking in order to fit in and/or to numb themselves. Interestingly, reasons like cost, using them to quit smoking, seeing them in the media, and ease of attainment were ranked the lowest when it came to smoking e-cigarettes. However, their lower costs naturally lead to increased accessibility compared to traditional cigarettes, making them easier to reach the hands of teenagers. Overall, in this paper, we aim to identify if adolescents are choosing e-cigarettes as their first foray into tobacco products and why.

## 1. Introduction

Since 2014, e-cigarettes have been the most frequently used tobacco product for teenagers in the United States (US) [1]. In 2018, the US Surgeon General announced that e-cigarette usage is becoming an epidemic amongst adolescents [2]. In 2022, the Centers for Disease Control and Prevention (CDC) discovered that 2.55 million middle and high school students were using electronic cigarettes (e-cigarettes) in the US alone [3]. E-cigarettes allow people to inhale a nicotine solution (e-liquid) into their lungs. The e-cigarette device accomplishes this by heating the e-liquid, enabling it to aerosolize and be subsequently inhaled [4]. These devices can take many forms, including but not limited to metered-dose inhalers, pipes, and cigars [5]. E-liquids generally contain a propylene glycol base, vegetable glycerin, flavorings, and up to 48 mg/mL of nicotine [6]. The fruit and candy flavors found in e-cigarettes increase their appeal amongst adolescents [7]. Additionally, flavor-related advertisements have increased adolescents’ interest in purchasing and using e-cigarettes [7].

While e-cigarettes are often advertised as a way to help people quit traditional tobacco products, the World Health Organization (WHO) has stated that there is no proof of e-cigarettes being effective at reducing an individual’s tobacco use [8]. Furthermore, despite e-cigarettes being advertised as a healthier alternative to traditional tobacco smoking, e-cigarettes are known to produce toxic compounds [5,8]. The use of e-cigarettes can increase an individual’s risk for lung burns and trauma, poisoning [1,9], addiction, and e-cigarette vaping-associated lung injury (EVALI) [4,9]. The condition EVALI is linked with vaping products with tetrahydrocannabinol (THC), the psychoactive component of marijuana, and Vitamin E acetate, a diluting agent for THC products [4,9]. Twenty-seven and a half percent of high school students used e-cigarettes in 2019 [10], and one-third of high school students that used e-cigarettes also used THC-containing products [9,11].

Overall, e-cigarette usage is becoming an epidemic, which can result in many negative health outcomes, among adolescents [2,12]. Therefore, it is important to know how many of our youth decide to smoke e-cigarettes over traditional cigarettes and why they smoke them. This will help us determine whether e-cigarettes are the first tobacco products being used by adolescents. Additionally, if we can determine what the most common predictors are for adolescent smoking, then we can develop preventative measures to limit e-cigarette incidence within the younger population.

All analyses were performed using data from the 2021 National Youth Tobacco Survey (NYTS). This survey is a cross-sectional study that is administered to US middle and high school students every year [13]. Three-stage cluster sampling was utilized to ensure that the sample of students was representative of the national student population [13]. Due to COVID-19 protocols, the 2021 NYTS was dispensed via a web URL, which allowed for in-person, virtual, and hybrid students to respond [13]. The protocol for the NYTS was approved by the Centers for Disease Control and Prevention (CDC)’s Institutional Review Board (IRB). Informed consent was obtained from both the respondents and their parents [14]. Our aim was to identify the reasons why middle and high school students are taking to vaping. We also aimed to discover sex and racial differences in e-cigarette smoking.

## 2. Materials and Methods

The student response rate for the 2021 NYTS was 81.2% and the school response rate was 54.9% [13]. The responses to the questions QN6 [Question Number 6] (“Have you ever used an e-cigarette, even once or twice?”) and QN35 [Question 35] (“Have you ever smoked a cigarette, even one or two puffs?”) formed the bedrock of our analyses. Using the responses, we identified students who smoked e-cigarettes exclusively, students who smoked cigarettes exclusively, and students who did both. See Table 1. Individuals were omitted from this study if they had no response for QN6, which was a “Yes” or “No” question. Overall, there were 19,955 student responses included in this analysis.

Demographic information comes from a different batch of questions. Information on sex comes from QN2 [Question Number 2], and race from QN4A, B, C, D, and E (Black, White, Asian, American or Alaska Native, and Hawaiian or Pacific Islander). Additionally, data on reasons why a subject first started smoking e-cigarettes (QN11A-M) were used.

To determine the age at which adolescents first started smoking, QN7 (“How old were you when you first used an e-cigarette, even once or twice?”) was used. Also, to determine mental health status, questions QN157A-D were analyzed.

All analyses were conducted using SAS v.9.4 software. A logistic regression analysis was used to estimate the odds ratio (OR) and a 95% confidence interval (CI) of e-cigarette smoking by race and sex when adjusted for demographic characteristics. One advantage of fitting a logistic regression model to the data is that it provides more interpretable results. Additionally, a Chi-squared analysis was employed when dealing with categorical variables.

## 3. Results

A.Incidences

A cross-tabulation of the answers to QN6 and QN35 is reported in Table 1 along with percentages.

The percentage of pupils who smoked cigarettes exclusively was 2.2%, whereas the percentage of pupils who vaped exclusively was 12.1%, which is 5.5 times higher. When the Chi-squared test of independence was used, the resulting *p*-value was less than 0.001. Therefore, adolescents are significantly more likely to smoke e-cigarettes than traditional cigarettes.

B.Demographics

The most visible demographics in the data were sex and race. We investigated whether sex and racial differences significantly manifested between those who vape and those who do not. In Table 2, we present our findings with cross-classified frequencies along with unadjusted *p*-values following the Chi-squared tests.

The numbers in Table 2 indicate that females were more likely to vape than males. The difference was 3.6 percentage points, which may seem to be close. However, the *p*-value is <0.001, making the difference significant. For race, among those who vaped, 7.7% were American Indian or Alaska Native; 5.0% were Asian; 17.2% were Black; 2.9% were Native Hawaiian or Other Pacific Islander; and 67.2% were White. Of the adolescents that did not use e-cigarettes, 6.4% were American Indian or Alaska Native; 7.4% were Asian; 23.6% were Black; 2.4% were Native Hawaiian or Other Pacific Islander; and 60.0% were White, as per the order of Table 2. The *p*-value between each race was <0.001, so there was a significant difference in the sample size between each race.

C.Logistic Regression

In the results presented in Table 2, there were no odds ratios and the *p*-values were unadjusted. The logistic regression model enabled us to discover differences between the sexes and races, adjusted in each case for the presence of the other. The findings are presented in Table 3. The response variables were the answers to QN6, which were binary with the levels “Yes” or “No” to the question “Have you ever used an e-cigarette, even once or twice?”. The predictors were sex and race.

Table 3 shows that females had significantly higher odds of smoking e-cigarettes than males (*p*-value =< 0.001) both before and after adjusting for race. Thus, females were positively associated with vaping (Table 3). For race, there was a significant difference in e-cigarette usage between White people and American/Alaska Native, Asian, Black, and Native Hawaiian/Pacific Islanders (Table 3).

D.Age Adolescents Start Vaping

Information on the age was obtained from QN7. The distribution of the ages at which pupils tried vaping for the first time is captured in Figure 1.

The bar graph depicts the age of each participant when they first smoked an e-cigarette (Figure 1). The ages with the highest frequency of smoking initiation were 13–17 years old. The peak was at 15 years old, the age when a pupil is typically in the ninth grade (Figure 1). Therefore, children seem to start testing out e-cigarette products when they are between the ages of 13 and 17.

E.Reason for Vaping

When a pupil responded “Yes” to QN6, he/she was directed to answer QN11 (“Why did you first use an e-cigarette?”). Thirteen reasons were offered. A student was able to check more than one reason. It was not easy to analyze such data. We provided simple summary statistics of the frequencies of the reasons. A bar chart was created in Figure 2, which depicts the percentage of pupils who responded to a reason listed in the figure.

In Figure 2, we see that females and males had similar reasons for starting to vape. The top two reasons, A and B, remained the same. The top reason was “A: A Friend Used Them”, which amounts to peer pressure. The second topmost reason was “B: I was curious about them”. Another topmost reason was “C: I was feeling anxious, stressed, or depressed”. There was virtually no support for the reason “L: To quit using other tobacco products”.

We also presented the numbers that comprise Figure 2.

We use Table 4 to clarify further the reasons for smoking e-cigarettes by sex. Both males and females used e-cigarettes because a friend used them (males = 22.4%; females= 22.87%) or they were curious about them (males = 19.18%; females = 19.46%) (Figure 2; Table 4). Males were 1.43 times more likely to smoke e-cigarettes than females because they perceived them to be less harmful than other forms of tobacco (*p*-value = 0.0103) (Figure 2; Table 4). On the other hand, females were 1.29 times more likely to smoke e-cigarettes than males due to feelings of anxiety, stress, and depression (*p*-value = <0.0001) (Figure 2; Table 4). Chi-squared tests revealed that there are significant differences between the sexes and the reasons that they vape. The only reasons that had no significant difference between the sexes were “To Quit Smoking Other Tobacco Products” (*p*-value = 0.0741), “Were Easier to Get Than Other Tobacco Products” (*p*-value = 0.923), and “People on TV, Online, or in Movies Use Them” (*p*-value = 0.296).

F.Mental Health Status

Mental health responses in the survey were identified by the answers to QN157A-D. Their responses, along with frequencies and percentages, are tabulated in Table 5 by sex.

Females were 1.57, 1.97, 2.40, and 2.42 times more likely than males to say that they had little pleasure in doing things, were feeling depressed or hopeless, were feeling nervous or anxious, and were unable to stop or control their worrying nearly every day (Table 5). On the other hand, males were more likely than females to say that they never experienced little pleasure in doing things (1.32), feelings of depression or hopelessness (1.45), feelings of anxiety or nervousness (1.69), and being unable to stop or control worrying (1.55) (Table 5).

## 4. Discussion

The two predominant ways that adolescents consume nicotine are by vaping and smoking traditional cigarettes. According to the data, the frequency of only e-cigarette smokers was 5.52 times the frequency of only traditional cigarette smokers (Table 1). It seems that vaping is the main vehicle by which a student ingests nicotine. Additionally, there was a statistically significant difference in the proportions of individuals who were only e-cigarette smokers, only traditional smokers, and both e-cigarette and traditional smokers (Table 1). This suggests that the first nicotine product that adolescents are being exposed to is primarily e-cigarettes. Furthermore, only 0.97% of e-cigarette smokers mentioned that their reason for vaping was to quit smoking (Figure 2). So, these individuals are not vaping to stop using tobacco products. Instead, e-cigarettes seem to be acting as their first foray into tobacco use. This goes against e-cigarette manufacturing companies, which advertise e-cigarettes as a way to quit smoking. In a comprehensive review, the neurological impacts of nicotine products on a developing, adolescent brain, and their long-term impacts on the brain, behavior, and addiction vulnerability, were evaluated [15]. The chemical changes that result from adolescent nicotine exposure could potentially explain why drug-related emergencies have increased at the same time as e-cigarette usage [15]. Furthermore, e-cigarette usage has been associated with marijuana [16] and combustible tobacco use [17,18], suggesting that the biochemical adaptations in the brain, due to nicotine use, may increase an adolescent’s risk of other drug addictions [19,20].

In the logistic regression analysis, neither sex nor race had a difference of 10% or more between the unadjusted and adjusted odds ratios, so these two variables did not seem to be confounding (Table 3). However, both sex and race had significant *p*-values. For sex, the odds ratio was greater than one, which indicates that females were significantly more likely to smoke e-cigarettes than males. In a previous study, females who never smoked were significantly less likely to vape than males who had never smoked [21]. While there seems to be a small difference in the odds of smoking between males and females, the data are large, so small differences end up being significant. Nevertheless, it has been hypothesized that stress is an important factor influencing traditional smoking in females [22]. This is supported by the results obtained from the NYTS. In the survey, females were twice as likely as males to feel down, depressed, or hopeless; feel nervous, anxious, or on edge; and be unable to stop or control worrying nearly every day (Table 5). While this study encompassed traditional smoking instead of e-cigarette smoking, a study on adolescents found that traditional and e-cigarette smokers exhibited similar behaviors [23]. Therefore, it is possible that mental health status could be a factor in why females had 1.172 times the adjusted odds of smoking e-cigarettes than males (Table 3). Furthermore, when the reasons for smoking e-cigarettes were broken down by biological sex, females were 1.29 times more likely than males to vape because they were feeling anxious, stressed, or depressed (Figure 2). This further validates that poor mental health may be contributing to females having higher odds of vaping than males.

Conversely, American/Alaska Natives, Asians, Blacks, and Native Hawaiian/Pacific Islanders all had lower e-cigarette usage than Whites because their odds ratios were less than one (Table 3). Only Asian, Black, and Native Hawaiian/Pacific Islanders had no significant difference in the odds of e-cigarette usage, as shown by their overlapping confidence intervals (Table 3). Furthermore, there was no significant difference in e-cigarette usage between Native Hawaiian/Pacific Islanders and American/Alaska Natives. However, there were not many Native Hawaiian/Pacific Islanders in this study, which could be the reason for the greater variance shown in the 95% confidence interval. White individuals had a significantly higher odds of smoking e-cigarettes than all of the other races (Table 3). This could be because Black and Hispanic individuals were shown to have more positive social norms surrounding traditional tobacco products compared to White individuals [24]. Positive attitudes about tobacco products in general had an inverse association with e-cigarette usage, since e-cigarettes are often marketed as a healthier alternative to traditional cigarettes [24]. This may explain why minorities had lower odds of vaping than White individuals.

Unsurprisingly, the main reasons for youth e-cigarette use (regardless of sex) were “a friend used them” (22.51%), “I was curious about them” (19.32%), and “I was feeling anxious, stressed, or depressed” (10.40%). Additionally, individuals typically started smoking e-cigarettes when they were between the ages of 13 and 17 years old (Figure 1). The age which individuals usually started smoking at was 15 years old (Figure 1). This is anticipated because over 30% of young adult smokers began smoking before they were 16 years old [25,26]. Without new smokers, the tobacco industry would be unable to sustain itself [25,26]. Therefore, many tobacco companies want to target the adolescent population [26]. An individual’s susceptibility to peer pressure typically starts in early adolescence and is shown to peak at age 14 [27]. Thus, using e-cigarettes because “a friend used them” makes sense given the age at which they started smoking. At this age, adolescents just want to fit in, so it is likely for them to vape if all their friends are doing it. Furthermore, younger individuals tend to be more curious and have a desire to have new experiences, leading to the use of e-cigarettes [28]. Lastly, mental health and substance use are often thought to be connected [29]. Both mental illness and drug use have similar biological risk factors. Additionally, individuals may use drugs to lessen their mental health symptoms. On the other hand, using drugs may also cause a mental illness to express or worsen [29].

While we believe that this analysis has provided useful information about the frequency of e-cigarette usage in adolescents and potential risk factors that increase the odds of vaping, the data used were only for one year, 2021. Therefore, we do not know if the proportion of e-cigarette users has increased over time or not. We also do not know if the reasons for e-cigarette use will change due to the increased normalization of vaping in the media. Thus, for future research, we should compare the reasons for smoking e-cigarettes in teens over multiple years to see if they change. Furthermore, it would be helpful to see if the proportion of adolescent e-cigarette users increases over time or not. In this paper, survey data were used. Thus, the responses from the participants cannot be verified independently.

Our analysis was based on survey data. As with any survey data, there is no way to check the accuracy of the data collected. That is a limitation.

## 5. Conclusions

The results from the NYTS showed that e-cigarettes were usually the first tobacco product used by adolescents, suggesting that vaping may act as a gateway to further drug use. The age group most at risk of smoking e-cigarettes were individuals aged 13–17, with 15 being the age when most individuals started vaping. This is important because a tobacco brand’s longevity is determined by how many younger individuals decide to smoke [26,30]. The reasons for smoking were because a friend smoked, they were curious, and/or mental health status. Overall, these results did not indicate that e-cigarettes were being used to quit traditional tobacco products amongst adolescents. Instead, they seemed to be enticing them to use more. This may be why countries such as Lebanon and Singapore have laws banning the importation, distribution, and sale of e-cigarettes [31]. However, knowing the main risk factors for vaping within the younger population will hopefully allow us to implement preventive measures, which will hopefully lower the incidence of vaping among adolescents. Continuing to monitor the prevalence of e-cigarette usage over time will also alert us to the presence of possible new risk factors, allowing us to find even more ways to reduce vaping incidence.

## Figures and Tables

**Figure 1 medicina-60-01541-f001:**
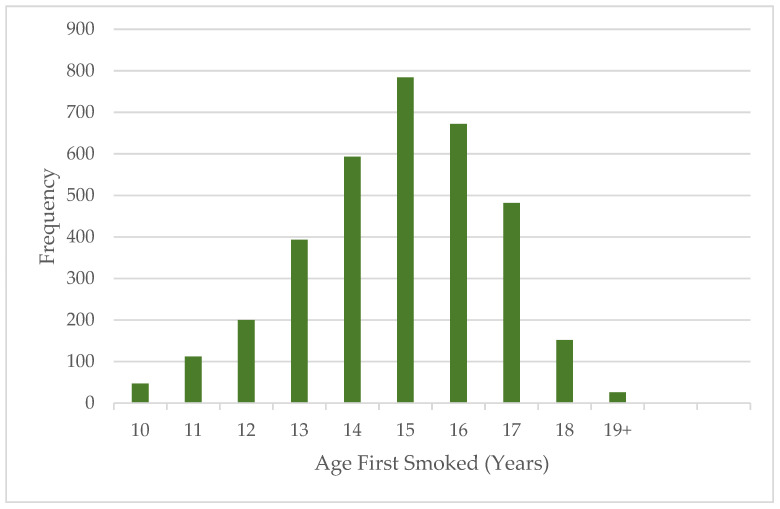
Age participants first started smoking e-cigarettes.

**Figure 2 medicina-60-01541-f002:**
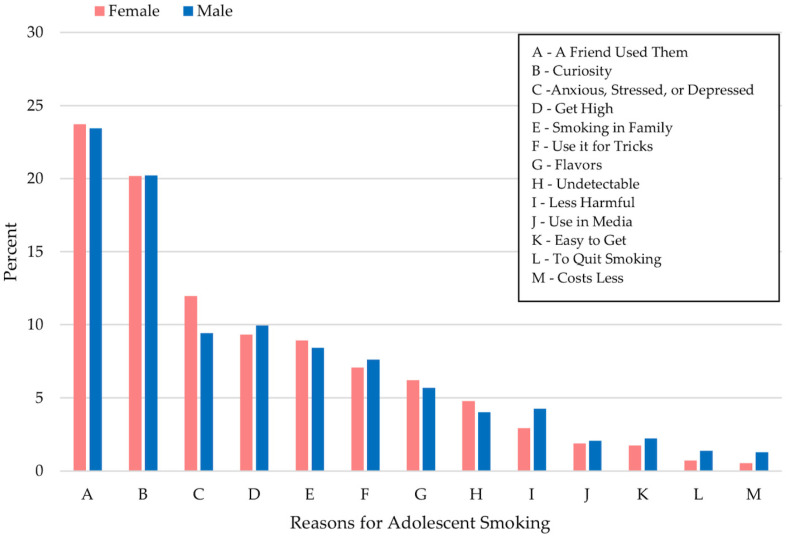
Reasons for vaping—percentages—by sex.

**Table 1 medicina-60-01541-t001:** Numbers of pupils cross-classified according to their status on e-cigarette and cigarette smoking.

	Cigarette		Total
E-Cigarette	No	Yes	
No	15,960 (80.1%)	437 (2.2%)	16,397
Yes	2414 (12.1%)	1119 (5.8%)	3533
Total	18,374	1556	19,930

**Table 2 medicina-60-01541-t002:** Demographics of pupils versus the status of vaping.

	E-Cigarette Yes	E-Cigarette No	*p*-Value
Sex			<0.001
Male	1757 (48.2%)	8519 (51.7%)	
Female	1891 (51.8%)	7956 (48.3%)	
Race			<0.001
American Indian or Alaska Native	303 (7.7%)	1075 (6.4%)	
Asian	197 (5.0%)	1231 (7.4%)	
Black	675 (17.2%)	3941 (23.6%)	
Native Hawaiian or Other Pacific Islander	112 (2.9%)	407 (2.4%)	
White	2632 (67.2%)	10,030 (60.0%)	

**Table 3 medicina-60-01541-t003:** Association between sex and vaping, and race and vaping with adjusted *p*-values and odds ratios.

Sociodemographic Characteristic	Unadjusted OR (95% CI)	Unadjusted *p*-Value	Adjusted OR (95% CI)	Adjusted *p*-Value
Sex				
Female	1.164 (1.083–1.253)	<0.001	1.172 (1.100–1.249)	<0.001
Race				
American/Alaska Native	0.872 (0.783–0.972)	0.013	0.875 (0.784–0.975)	0.016
Asian	0.647 (0.575–0.727)	<0.001	0.645 (0.573–0.726)	<0.001
Black	0.667 (0.613–0.726)	<0.001	0.667 (0.613–0.726)	<0.001
Hawaiian/Pacific Islander	0.772 (0.676–0.882)	<0.001	0.774 (0.677–0.885)	<0.001

**Table 4 medicina-60-01541-t004:** Reasons for smoking based on biological sex.

Reasons for Smoking E-Cigarettes	Frequency Males (%)	Frequency Females (%)	*p*-Value
A Friend Used Them	865 (22.24)	1088 (22.87)	<0.0001
A Family Member Used Them	311 (7.99)	409 (8.60)	<0.0001
To Quit Smoking Other Tobacco Products	51 (1.31)	33 (0.69)	0.0741
Cost Less Than Other Tobacco Products	47 (1.21)	24 (0.50)	0.0103
Were Easier to Get Than Other Tobacco Products	82 (2.11)	80 (1.68)	0.9233
People on TV, Online, or in Movies Use Them	76 (1.95)	86 (1.81)	0.2969
They Are Less Harmful Than Other Forms of Tobacco	157 (4.04)	134 (2.82)	0.0103
They Are Available in Flavors	210 (5.40)	285 (5.99)	0.0001
Could Be Used Unnoticed at Home or at School	148 (3.80)	219 (4.60)	<0.0001
I Could Use Them to do Tricks	281 (7.22)	325 (6.83)	0.0207
I Was Curious About Them	746 (19.18)	926 (19.46)	<0.0001
I Was Feeling Anxious, Stressed, or Depressed	348 (8.95)	549 (11.54)	<0.0001
To Get a High or Buzz from Nicotine	367 (9.43)	428 (9.00)	0.0054

**Table 5 medicina-60-01541-t005:** Mental health status by sex.

Reasons for Smoking E-Cigarettes	Frequency Males (%)	Frequency Females (%)	*p*-Value
**Little Pleasure in Doing Things**			<0.0001
Not at All	5476 (62.2)	4132 (47.15)	
Several Days	1852 (20.93)	2322 (26.5)	
More Than Half Days	699 (7.94)	1081 (12.34)	
Nearly Every Day	784 (8.91)	1228 (14.01)	
**Feeling Down, Depressed, or Hopeless**			<0.0001
Not at All	5693 (64.95)	3922 (44.73)	
Several Days	1757 (20.05)	2342 (26.71)	
More Than Half Days	623 (7.11)	1140 (13.00)	
Nearly Every Day	692 (7.90)	1364 (15.56)	
**Feeling Nervous, Anxious, or on Edge**			<0.0001
Not at All	5088 (57.99)	2996 (34.26)	
Several Days	2107 (24.01)	2429 (27.77)	
More Than Half Days	782 (8.91)	1415 (16.18)	
Nearly Every Day	797 (9.08)	1906 (21.79)	
**Not Being Able to Stop or** **Control Worrying**			<0.0001
Not at All	6100 (69.72)	3924 (44.89)	
Several Days	1454 (16.62)	2065 (23.62)	
More Than Half Days	536 (6.13)	1161 (13.28)	
Nearly Every Day	659 (7.53)	1592 (18.21)	

## Data Availability

The original data presented in the study are openly available on CDC website: About National Youth Tobacco Survey (NYTS)|Smoking and Tobacco Use|CDC.

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
