# Peer review of "Vaping: The Key to Quitting Cigarettes or a Gateway to Addiction?"

_medicina, 2024, doi:10.3390/medicina60091541_

Round 1
Reviewer 1 Report
Comments and Suggestions for Authors
Overall, the paper is well-structured and presents its arguments effectively. The research is thorough, and the writing is clear and concise, making it accessible to a wide audience. However, I have few specific observations that I believe could enhance the quality of the paper.
1. Line no 75: The protocol for NYTS was approved by the Centers for Disease Control and Prevention’s (CDC) Institutional Review Board (IRB). Informed consent was obtained from both the respondents and their parents”
This statement is not essential as it is not the methodology of your study; instead, it is how the NYTS is administered.
2. Several countries have banned e-cigarettes due to concerns that they may act as a gateway to smoking. Highlighting these examples could strengthen your discussion
3. To enhance clarity for the reader, it would be helpful to specify the dependent and independent variables in the study.
Author Response
Dear Reviewer, Thank you for all your time and suggestions. Please see my point- to -point responses to your comments:
Comments and Suggestions for Authors
Overall, the paper is well-structured and presents its arguments effectively. The research is thorough, and the writing is clear and concise, making it accessible to a wide audience. However, I have few specific observations that I believe could enhance the quality of the paper.
Comment 1. Line no 75: The protocol for NYTS was approved by the Centers for Disease Control and Prevention’s (CDC) Institutional Review Board (IRB). Informed consent was obtained from both the respondents and their parents”
This statement is not essential as it is not the methodology of your study; instead, it is how the NYTS is administered.
Response 1: The NYTS administration process is important because the data we are using is based on this survey. It shows that this survey has the proper approvals and validates the data used in the study. IRB approval is required by many biomedical journals. This is the reason we discuss the IRB in the paper. We put this IRB discussion to Introduction (Line 72-80). Thank you for the suggestion.
Comment 2: Several countries have banned e-cigarettes due to concerns that they may act as a gateway to smoking. Highlighting these examples could strengthen your discussion
Response 2: Added new reference (Line 556-557) and an example (Line 410-411). Thank you and please review.
Comment 3: To enhance clarity for the reader, it would be helpful to specify the dependent and independent variables in the study.
Response 3: Thank you for your suggestion. We identified clearly the response variable and predictors in “C. Logistic Regression” (lines 166 to 168). Please review.
Best wishes,
Tianyuan Guan
Reviewer 2 Report
Comments and Suggestions for Authors
The aim of this study was to determine reasons for which adolescents start to smoke e-cigarettes. This is a very important public heath concern. The main strength of the study is the design and using representative NYTS data. However, the statitsical analysis and the quality of the manuscript must be improved. Here are my suggenstions and comments:
line 30 - did you mean "leads" instead of "lends"?
The Introduction is missing the aim of the study.
The Materials and Methods feels incomplete.
The number of surveys/students included in the study is missing.
line 107: "It is puzzling why pupils have taken to vaping. There must be other reasons. One of the aims is to find out why." The language is informal, and the aim is in Results - this must be corrected.
lines 120-125 - these results are described insufficiently, and some comments belong rather to Discussion (eg "Since the data is large, small differences turn out to be significant")
Please provide summary information on the studied group: mean/median age, sex distribution, race distribution etc.
line 140 - "One advantage in fitting a logistic regression model to the data is that it provides more interpretable results." Again, such a sentence belongs either to Materials and Methods or Discussion.
line 143 - the first two sentences repeat the same information
Figure 1 - Age is a continuous variable, this should be a histogram, not a bar plot. Also, showing the relative frequency of the y-axi would be more informative.
line 150 - Results mention QN7 although it was not described in Materials and Methods.
line 162 - as above, QN12 not described in Materials and Methods.
Figure 3 basically repeats the information from Figure 2, so one of them seems to be redundant.
lina 174 - What statistical procedure was performed to establish the "remarkable agreement"?
lines 178-180 - the last two sentences of this paragraph belong to Discussion
line 216 - there is no mention of investigating mental health status in Materials and Methods
Results from Tables 4 and 5 - when talking about one group being more or less likely of something than the other, please provide results of statistical procedures confirming these statements. Otherwise, you provide only descriptive statistics (frequencies) from which you cannot draw any conclusions.
Discussion - the whole section needs rewriting according to the corrected statistical analysis. It would be a good idea to use subsections that focus on one specific aspect studied.
From line 362 - make sure to delete any text from the article template such as "6. Patents This section is not mandatory but may be added if there are patents resulting from the work reported in this manuscript." etc.
Please provide the whole questionnaire as supplementary material.
Author Response
Dear Reviewer, Thank you for all your time and suggestions. Please see my point- to -point responses to your comments:
Comments and Suggestions for Authors
The aim of this study was to determine reasons for which adolescents start to smoke e-cigarettes. This is a very important public health concern. The main strength of the study is the design and using representative NYTS data. However, the statistical analysis and the quality of the manuscript must be improved. Here are my suggestions and comments:
Comment 1: line 30 - did you mean "leads" instead of "lends"?
Response 1: Thank you for the suggestion. I fixed it in line 29-31.
Comment 2: The Introduction is missing the aim of the study.
Response 2: We added “aim” to the abstract and Introduction. (lines 32-24 and 82-83)
Comment 3: The Materials and Methods feels incomplete.
Response 3: We added more content to the Methods section. (lines 100-102, 109-111, 117-119)
Comment 4: The number of surveys/students included in the study is missing.
Response 4: Number of distinct students included in the survey analysis was 19,955. (line 102)
Comment 5: line 107: "It is puzzling why pupils have taken to vaping. There must be other reasons. One of the aims is to find out why." The language is informal, and the aim is in Results - this must be corrected.
Response 5: Addressed. Lines 132-135. Please review.
Comment 6: lines 120-125 - these results are described insufficiently, and some comments belong rather to Discussion (eg "Since the data is large, small differences turn out to be significant")
Response 6: Addressed. Lines 339-341
Comment 7: Please provide summary information on the studied group: mean/median age, sex distribution, race distribution etc.
Response 7: See Table 1 (line 128-131) and 2 (line 146-147) for descriptive statistics.
Comment 8: line 140 - "One advantage in fitting a logistic regression model to the data is that it provides more interpretable results." Again, such a sentence belongs either to Materials and Methods or Discussion.
Response 8: Addressed. Lines 117-119.
Comment 10: line 143 - the first two sentences repeat the same information
Response 10: Addressed. Lines 147-150.
Comment 11: Figure 1 - Age is a continuous variable, this should be a histogram, not a bar plot. Also, showing the relative frequency of the y-axis would be more informative.
Response 11: I appreciate the comment about using a histogram instead of a bar chart. The Age First Smoked is a discrete variable. For QN7 “How old were you when you first used an e-cigarette, even once or twice?”, the possibilities were 01 (8 years old or younger), 02 (9 yrs), 03 (10 yrs), 04 (11 yrs), 05 (12 yrs), 06 (13 yrs), 07 (14 yrs), 08 (15 yrs), 09 (16 yrs), 10 (17 yrs), 11 (18 yrs), 12 (19 years or older).
Comment 12: line 150 - Results mention QN7 although it was not described in Materials and Methods.
Response 12: Addressed. Line 109-111.
Comment 13: line 162 - as above, QN12 not described in Materials and Methods.
Response 13: This was supposed to be QN11A-M, which was mentioned in the Materials and Methods. QN11 is the reason why the student started smoking. QN12 is why the student continues to smoke. QN12 will be added to the supplemental material.
Comment 14: Figure 3 basically repeats the information from Figure 2, so one of them seems to be redundant.
Response 14: Thank you for this suggestion. We deleted Figure 2. Line 206-210.
Comment 15: line 174 - What statistical procedure was performed to establish the "remarkable agreement"?
Response 15: Chi-squared test was used. Line 259.
Comment 16: lines 178-180 - the last two sentences of this paragraph belong to Discussion
Response 16: Addressed. Line 324-325.
Comment 17: line 216 - there is no mention of investigating mental health status in Materials and Methods
Response 17: Addressed. Line 110-111.
Comment 18: Results from Tables 4 and 5 - when talking about one group being more or less likely of something than the other, please provide results of statistical procedures confirming these statements. Otherwise, you provide only descriptive statistics (frequencies) from which you cannot draw any conclusions.
Response 18: Thank you for your suggestion we ran a Chi-squared test for tables 4 and 5. However, the Chi-squared test for Table 5 is not very informative because there is a large difference in value between the different levels for each question. The following four reasons stand out for taking to vaping. 1. Peer Pressure – A Friend Used Them (1753/8275 = 21.2% sexes combined); 2. Curiosity (1672/8275 = 20.2% sexes combined); 3. Feeling Anxious, Distressed, or Depressed – 897/8275 = 10.8% sexed combined); 4. Getting High – 795/8275 = 9.6%). The reason that vaping helps quit other products gets a short shrift. Only 84/8275 (= 1.0%) gave this reason for vaping. It ranks 12th of all the reasons offered. However, the numbers are small. Additionally, males offer the reason that vaping helps quitting other products significantly more so than females (1.31% versus 0.69%). However, the numbers are small. One should look at other year’s data for corroboration. Please review our new Table 4 and 5.
Comment 19: Discussion - the whole section needs rewriting according to the corrected statistical analysis. It would be a good idea to use subsections that focus on one specific aspect studied.
Response 19: Addressed. Line 324-325, and 339-341.
Comment 20: From line 362 - make sure to delete any text from the article template such as "6. Patents This section is not mandatory but may be added if there are patents resulting from the work reported in this manuscript." etc.
Response 20: Addressed, thank you.
Comment 21: Please provide the whole questionnaire as supplementary material.
Response 21: The questionnaire is too long to be included in the supplement. We are providing the linking to the questionnaire and datasets in the “Data Availability Statement” section.
(https://www.cdc.gov/tobacco/about-data/surveys/historical-nyts-data-and-documentation.html.)
Thank you again and best wishes,
Tianyuan Guan
Reviewer 3 Report
Comments and Suggestions for Authors
The debate over whether e-cigarettes help with smoking cessation or serve as a gateway to addiction continues, making research that can answer this question particularly meaningful. However, the analysis presented in the submitted manuscript must convincingly support the conclusions drawn. Therefore, I believe the following aspects need to be addressed:
-
The introduction should persuasively present the rationale that the research question, as indicated in the title, can be tested using descriptive statistics. While this study analyzes adolescent e-cigarette use, it does not seem to convincingly support the validity of the conclusions drawn from it.
-
Additionally, the conclusions presented in the abstract and main body appear to be somewhat disconnected from the research question stated in the title.
-
When presenting numerical data, it is advisable to use commas as thousand separators for numbers with four digits or more.
-
What is the purpose of including mental health status in this study?
-
There seem to be more limitations to this study that have not been addressed. Please provide a more detailed explanation.
Author Response
Dear Reviewer, Thank you for all your time and suggestions. Please see my point- to -point responses to your comments:
Comments and Suggestions for Author
The debate over whether e-cigarettes help with smoking cessation or serve as a gateway to addiction continues, making research that can answer this question particularly meaningful. However, the analysis presented in the submitted manuscript must convincingly support the conclusions drawn. Therefore, I believe the following aspects need to be addressed:
Comment 1: The introduction should persuasively present the rationale that the research question, as indicated in the title, can be tested using descriptive statistics. While this study analyzes adolescent e-cigarette use, it does not seem to convincingly support the validity of the conclusions drawn from it.
Response 1: We hit upon the title after analyzing the data. There are five takeaways from our analysis:
1. Among middle and high school students, there are significantly more students vaping exclusively than students smoking exclusively (12.1% vs. 2.1%).
2. The third most predominate reason for taking to vaping is anxiety, stress, or depression.
3. Differences between the sexes is significant on this count with females taking to vaping more likely than males.
4. Manufacturers of vaping products are advocating the use of e-cigarettes to quit smoking. There is not happening as per the data. This point is highlighted in the title.
5. Whether one vapes or smokes, the adolescent is exposed to nicotine, which is addictive. This is pointed out in the title.
Comment 2: Additionally, the conclusions presented in the abstract and main body appear to be somewhat disconnected from the research question stated in the title.
Response 2: Addressed. Please review the new abstract.
Comment 3: When presenting numerical data, it is advisable to use commas as thousand separators for numbers with four digits or more.
Response 3: Found no instances where a comma wasn’t used to separate numbers with four digits or more. Thank you.
Comment 4: What is the purpose of including mental health status in this study?
Response 4: Mental health was the third most prevalent reason discovered in this paper for starting to vape. It was also the risk factor with the greatest difference between males and females.
Comment 5: There seem to be more limitations to this study that have not been addressed. Please provide a more detailed explanation.
Response 5: Thank you for this suggestion, we added more limitations in the discussion section. Please review.
Thank you and best wishes,
Tianyuan Guan
Round 2
Reviewer 2 Report
Comments and Suggestions for Authors
Although some revisions have been made by the authors, the manuscript is still not ready for publication. I cannot find the author's changes as the lines indicated in author's reply do not match lines in the text.
Author Response
See the attached response to Reviewer 2.
